# Molecular Epidemiology, Antimicrobial Resistance, and Clinical Characteristics of *Streptococcus pneumoniae* Isolated from Adult Patients with Invasive Pneumococcal Disease

**DOI:** 10.3390/antibiotics14111158

**Published:** 2025-11-15

**Authors:** Kristina Franjić Amančić, Bojana Mohar-Vitezić, Đurđica Cekinović Grbeša, Tanja Grubić Kezele, Maja Abram, Marina Bubonja-Šonje

**Affiliations:** 1Department of Clinical Microbiology, Clinical Hospital Centre Rijeka, 51000 Rijeka, Croatia; kristina.franjic@uniri.hr (K.F.A.); bojana.mohar@uniri.hr (B.M.-V.); tanja.grubic@uniri.hr (T.G.K.); maja.abram@uniri.hr (M.A.); 2Department of Microbiology and Parasitology, Faculty of Medicine, University of Rijeka, 51000 Rijeka, Croatia; 3Clinic for Infectious Diseases, Clinical Hospital Centre Rijeka, 51000 Rijeka, Croatia; durdica.cekinovic@uniri.hr; 4Department of Infectious Diseases, Faculty of Medicine, University of Rijeka, 51000 Rijeka, Croatia; 5Department of Basic Medical Sciences, Faculty of Health Studies, University of Rijeka, 51000 Rijeka, Croatia

**Keywords:** antimicrobial resistance, invasive pneumococcal disease (IPD), MLST, penicillin, serotype, *Streptococcus pneumoniae*

## Abstract

**Background/Objectives**: *Streptococcus pneumoniae* is a major human pathogen causing illnesses that range from mild respiratory infections to severe invasive diseases. More than 100 known *S. pneumoniae* serotypes differ in their virulence, prevalence, and levels of drug resistance. Additionally, different clonal types within the same serotype may exhibit varying disease potential and genetic characteristics. This study aimed to determine phenotypic and molecular characteristics of *S. pneumoniae* isolated from patients with invasive pneumococcal disease (IPD). **Methods**: The serotypes of invasive *S. pneumoniae* isolates collected between 2022 and 2025 from adult patients hospitalized in a tertiary hospital were determined. Multilocus sequence typing (MLST) was performed on isolates with reduced susceptibility to penicillin to assess their molecular epidemiology. **Results**: Serotype 3 was the most common among all invasive isolates (29/85; 34.1%), followed by serotype 19A (22/85; 25.9%). Most penicillin-resistant isolates belonged to serotypes 19A and 19F. Three of the eight 19A isolates with reduced penicillin susceptibility were assigned to ST320 (37.5%), a clinically significant clone due to its high virulence and antibiotic resistance. While 15.3% of all isolates were multidrug-resistant (MDR), nearly half of the isolates with reduced penicillin susceptibility were MDR, most frequently exhibiting the erythromycin–clindamycin–tetracycline resistotype. **Conclusions**: This study highlights the predominance of serotype 19A, particularly the highly virulent and resistant ST320 clone, among invasive isolates with reduced penicillin susceptibility. These findings underscore the ongoing threat of antimicrobial resistance in IPD and the importance of continued surveillance of serotype distribution and resistance patterns to guide treatment strategies and vaccination policy decisions.

## 1. Introduction

*Streptococcus pneumoniae* is a leading cause of various clinical syndromes, including pneumonia, otitis media, sinusitis, and invasive pneumococcal diseases (IPD) such as sepsis, meningitis, and endocarditis. Although it is part of the normal nasopharyngeal microbiota, under certain conditions *S. pneumoniae* can become pathogenic, particularly in older adults and individuals with pre-existing conditions. Classification of *S. pneumoniae* into serotypes is based on differences in the capsular polysaccharide, a major determinant of virulence and vaccine effectiveness. Numerous polysaccharide and conjugate vaccines targeting the capsule are in use worldwide [1]. In Croatia, the 10-valent pneumococcal conjugate vaccine (PCV10) was introduced into the national childhood immunization program in 2019, while the national public health authority began recommending the 23-valent pneumococcal polysaccharide vaccine (PPV23) for older adults in 2021 [2]. Studies from various European countries with a longer history of pneumococcal vaccination have shown that the introduction of childhood immunization programs has led to a marked reduction in IPD cases among infants and, consequently, among adults [3,4,5].

Of particular concern is the increasing resistance of *S. pneumoniae* to penicillin, which has long been the antibiotic of choice for treating pneumococcal infections. Moreover, *S. pneumoniae* is showing rising resistance to other commonly used antibiotics, such as macrolides, lincosamides, and cephalosporins [6,7]. Multidrug resistance (MDR) in invasive *S. pneumoniae* strains is a growing global health concern due to its negative impact on treatment efficacy, particularly in serious infections [8]. This increasing resistance limits the effectiveness of standard therapies and underscores the need for ongoing surveillance. As antimicrobial susceptibility varies by geographic region and site of infection [5], treatment guidelines should be based on local epidemiological data.

Multilocus sequence typing (MLST) remains a powerful and widely used method for characterizing bacterial strains. In the case of *S. pneumoniae*, it assigns sequence types (STs) to isolates based on the allelic profiles of seven housekeeping genes, enabling comparison of strains across different time periods and geographic regions. However, only a limited number of specific *S. pneumoniae* STs from Croatian isolates have been reported in publicly available databases, with most data dating back over two decades ago [9].

The aim of this study was to assess the clinical and microbiological features of IPD in adults from western Croatia during the early period of PCV10 introduction, with particular attention to the molecular analysis of *S. pneumoniae* isolates showing reduced susceptibility to penicillin.

## 2. Results

### 2.1. Clinical–Demographic Characteristics of IPD Patients

Throughout the study period a total of 85 invasive *S. pneumoniae* isolates were collected from hospitalized adult patients. One isolate per patient was identified and tested. Most of the isolates were from elderly patients aged ≥ 65 years (60%), followed by adults aged 50–64 years (29.4%) (Table 1).

None of the patients (mean age: 68.4 years) had received pneumococcal vaccination. Males were slightly more prevalent among IPD patients (46 cases; 54.1%), although the difference was not statistically significant. A significant majority of IPD occurred in individuals with risk factors for pneumococcal infection, such as having two or more comorbidities (55 cases; 64.7%; *p* < 0.001). In the analysis of comorbidities across all studied cases, hypertension was the most prevalent condition, affecting 33 patients (38.8%), followed by malignancy in 18 (21.2%), diabetes mellitus in 17 (20%) and heart failure in 16 patients (18.8%). Among all patients, only 9 of them (10.6%) had no additional diseases. One-third of the IPD patients were admitted to the intensive care unit (ICU) (33 cases; 38.8%). The median estimated 10-year survival rate, based on a Charlson Comorbidity Index (CCI) score of 4, was 53%. The observed mortality rate among adults with IPD (25 cases; 29.4%) was statistically significantly higher than the average annual mortality rate in the adult Croatian population (*p* < 0.001) [10]. Blood was the most common specimen source, accounting for 87.1% of samples (74 cases; *p* < 0.001), followed by cerebrospinal fluid (CSF) in 6 cases. Other specimens, including pleural effusion, abscess aspirate, and joint aspirate, were less frequently observed (Table 2).

Comparison of surviving and deceased IPD patient groups revealed that age significantly affected survival, with deceased patients being older (mean 74.2 years) than survivors (mean 66.1 years; *p* = 0.029). Although a slightly higher proportion of women survived, the difference was not statistically significant. Most deceased patients (84%) had two or more comorbidities, but the overall difference in underlying conditions between groups was not significant. ICU admissions were significantly higher among deceased patients (*p* < 0.001). A CCI score of 5 in the deceased group corresponded to a significantly reduced estimated 10-year survival rate of 21% (*p* = 0.010), compared to 53% for a CCI score of 4 in survivors (Table 3).

### 2.2. Antimicrobial Susceptibility Testing

Antimicrobial susceptibility testing of invasive *S. pneumoniae* isolates showed that only 52 (61.2%) were fully susceptible to penicillin. Isolates with increased exposure penicillin susceptibility (I category) accounted for 27% (23/85), while 11.8% (10/85) were classified as penicillin-resistant (Figure 1).

The susceptibility of invasive *S. pneumoniae* isolates to penicillin, ceftriaxone, and meropenem—β-lactam antibiotics commonly used for IPD treatment—was assessed. The minimum inhibitory concentration (MIC)_90_ for penicillin was 0.75 mg/L, whereas ceftriaxone and meropenem exhibited lower MIC_90_ values of 0.5 mg/L and 0.094 mg/L, respectively (Table 4).

Resistance to erythromycin was observed in 15/85 strains (17.7%), to moxifloxacin in 5/85 strains (5.9%), to tetracycline in 13/85 strains (15.2%), and to trimethoprim-sulfamethoxazole in 17/85 (20%) strains. All strains were susceptible to vancomycin (Table 5). Regarding MDR resistance, 13/85 strains (15.3%) were classified as MDR.

### 2.3. Serotype Distribution

Among 23 detected *S. pneumoniae* serotypes, the most prevalent, in descending order, were serotypes 3, 19A, 8, 15A, 14 and 4 (Figure 2). Three isolates were non-typeable, likely due to low capsule production. The majority of IPD cases (83.5%) were caused by serotypes not included in PCV10 vaccine. The most common non-PCV10 serotypes were 3 (29/85; 34.1%) and 19A (22/85; 25.9%), which together accounted for 60% of all cases. However, serotypes not covered by PCV13 and PPV23—the vaccines recommended for individuals aged 65 and older—accounted for only 15.3% of cases. These included serotypes 6C, 9A, 15A, 15C, 18A, 23A, 23B, 24F, 25A, 28A and 31, each isolated from a single patient, except for serotype 15A, which was isolated from three patients.

We further analyzed the distribution of *S. pneumoniae* serotypes with reduced susceptibility to penicillin (Figure 3). Isolates with increased-exposure penicillin susceptibility (I category) were mostly associated with serotypes 3 (7/23; 30.4%) and 19A (6/23; 26.1%). Most penicillin-resistant isolates (R category) belonged to serotypes 19A (4/10; 40%) and 19F (2/10; 20%). One penicillin-resistant strain could not be typed.

### 2.4. Multilocus Sequence Typing Analysis

Given the clinical significance of penicillin non-susceptible *S. pneumoniae*, MLST was performed on 20 isolates with reduced penicillin susceptibility. These isolates, representing distinct serotypes, revealed a wide range of sequence types (STs) (Figure 4). Some STs were associated with specific serotypes, while others showed serotype heterogeneity. ST320 was associated with serotype 19A (*n* = 3), ST205 with serotype 4, and ST1377 with serotype 3. Notably, ST199 was associated with two serotypes (19A and 18A). Serotype 19A displayed the greatest diversity of STs, including ST199, ST319, ST320, ST416, ST994, and ST4467.

We further analyzed *S. pneumoniae* isolates with reduced susceptibility to penicillin according to their sequence types, serotypes and susceptibility to non-β-lactam classes of antibiotics (Table 6). Nearly half of these isolates were MDR, most commonly resistant to erythromycin, clindamycin, and tetracycline. All three ST320 isolates, associated with serotype 19A, exhibited this MDR resistotype and originated from residents of the same nursing home, suggesting a likely epidemiological link. Sequence types ST320, ST1830, ST271, ST319, and ST4467 were resistant to nearly all non-β-lactam classes, whereas ST199, associated with serotypes 19A and 18A, remained susceptible to all non-β-lactam antimicrobial classes.

## 3. Discussion

This study investigated the molecular epidemiology, antimicrobial resistance, and clinical characteristics of *S. pneumoniae* isolates from adult patients with invasive pneumococcal disease. By examining circulating strains, their resistance patterns, and associated clinical risk factors, we provide insights into the epidemiology of IPD that are relevant for both patient management and public health.

### 3.1. Clinical Characteristics and Outcomes

Age is a well-established risk factor for IPD. In our study, most cases occurred in elderly individuals with additional risk factors, such as the presence of two or more underlying comorbidities. The mean age of our patients was 68.4 years. Most subjects (60%) were older than 65 years, and 90% had at least one risk factor for developing IPD. A similar finding was reported in a 2025 study conducted among adults in France, where more than half of the individuals were over 65 years of age, and 70% had at least one risk factor [11]. Notably, none of our patients had received any pneumococcal vaccines, as the Croatian national immunization program does not include mandatory adult pneumococcal vaccination, and vaccine coverage in this population remains very low [12].

Clinical outcomes of pneumococcal infections are influenced by several factors, including the presence of comorbidities and the invasiveness of the infecting serotype—both of which are associated with poor prognosis [13]. In our study, the most common comorbidity among patients was hypertension, followed by malignancy, diabetes mellitus, and heart failure. A recent systematic review identified the highest risk of IPD among adults with immunocompromising conditions, such as organ transplantation, asplenia, and hematologic malignancies [14]. While advancing age is associated with a moderate increase in IPD risk, underlying comorbidities appear to be more significant contributing factors. In our cohort, older age and comorbidities were common findings, particularly among non-survivors. The severity of IPD was underscored by the fact that more than one-third of patients required ICU admission. The observed mortality rate in our study was 29.4%, consistent with global data, where IPD mortality ranges from 11% to 30% and can reach up to 55% in elderly or comorbid populations [15].

Bacteremic pneumonia was the most frequent clinical presentation of IPD in our patients (87.1%), while meningitis occurred in 7.1%. Septic arthritis, peritonitis, necrotizing fasciitis, and abscesses were observed in a few cases. In adults, IPD typically presents as pneumonia with bacteremia and is caused by a limited number of serotypes, the distribution of which varies by region, age group, and population.

### 3.2. β-Lactam Susceptibility

β-lactam antibiotics remain the first-line treatment for pneumococcal disease; however, penicillin non-susceptibility among invasive *S. pneumoniae* isolates poses a growing challenge, with higher prevalence in southern and eastern Europe. In 2022, 12% of European countries reported to the European Centre for Disease Prevention and Control (ECDC) that over 25% of IPD strains exhibited reduced penicillin susceptibility [5].

In this study, 27% of IPD isolates were classified as susceptible at increased exposure (I category), 11.8% were penicillin-resistant, and 61.2% were fully susceptible, reflecting an upward trend in Croatia compared to 2022 data (18% I category, <1% resistant) [16]. Serotype 3 dominated overall (29/85; 34.1%), while I-category isolates were mainly serotypes 3 (30.4%) and 19A (26.1%). Penicillin-resistant isolates primarily comprised serotypes 19A (4/10; 40%) and 19F (2/10; 20%), both clinically significant due to their association with antimicrobial resistance. Serotype 19A is highly invasive and frequently linked to post-vaccine serotype replacement in PCV10 using countries [17]. Although less invasive, serotype 19F often remains multidrug- resistant despite coverage by both PCV10 and PCV13 [18].

The penicillin MIC_90_ of our isolates was 0.75 mg/L, indicating moderate reduced susceptibility, particularly relevant for non-meningitis infections with higher clinical breakpoints. Similarly, a Spanish study conducted during the period 2007–2021 reported the same MIC_90_ value [7]. The ceftriaxone MIC_90_ of 0.5 mg/L in our isolates remains within the susceptible range for most clinical indications, although continued monitoring is warranted given rising resistance. European, and Croatian surveillance data have consistently shown very high susceptibility to ceftriaxone over many years, with only a few isolates demonstrating resistance or requiring increased exposure [5,15,19].

Meropenem demonstrated strong potency, with an MIC_90_ of 0.094 mg/L, supporting its use as empirical therapy for severe infections. However, resistance may gradually emerge as combined penicillin-binding proteins (PBPs) mutations can raise MICs to ≥0.25 mg/L [20].

These findings underscore the critical role of local antimicrobial surveillance in guiding empirical therapy, particularly in the context of evolving resistance patterns among invasive *S. pneumoniae* strains.

### 3.3. Molecular Epidemiology and Serotype Distribution

Monitoring temporal and regional trends in serotype distribution is essential for detecting serotype replacement and assessing vaccine effectiveness. Pre-PCV10 surveillance in Croatia (2005–2019) identified serotypes 3, 14, 19A, 9V, 7F, and 23F as the main causes of adult IPD [16], while 2022 ECDC data showed serotypes 3, 8, 19A, 22F, and 6C as most prevalent across Europe [19]. In our study (2020–2025, early PCV10 era), 83.5% of IPD cases were caused by non-PCV10 serotypes, with serotypes 3 and 19A dominating (60% of cases). Only a few cases involved vaccine-covered serotypes (4, 14, 19F), and previously common serotypes such as 9V and 7F were absent, reflecting typical post-vaccine serotype replacement.

These findings align with other post-introduction surveillance studies reporting increased incidence of serotypes 3 and 19 in countries using PCV10-only programs [21,22]. Similarly, Krajcar et al. recently reported dominance of these serotypes among Croatian children with community-acquired pneumonia [23], emphasizing the need for higher-valence vaccines, such as PCV13, to cover serotypes 3 and 19 in Croatia.

MLST has been widely used to study the evolution and global spread of virulent *S. pneumoniae* clones. However, such data are currently lacking for Croatia. In this study, MLST of 20 *S. pneumoniae* isolates from IPD cases with reduced penicillin susceptibility, collected in 2024, revealed substantial genetic heterogeneity, identifying 15 distinct sequence types (STs). ST320 associated with serotype 19A, was the most frequent, representing 15% of isolates. Notably, three ST320/19A isolates originated from residents of the same nursing home, suggesting a possible epidemiological link.

ST320 has been widely reported in other regions, particularly in association with serotype 19A and multidrug resistance. In the Asian Network for Surveillance of Resistant Pathogens (ANSORP) study, ST320 accounted for over 50% of serotype 19A isolates, reflecting its rapid expansion in the post-PCV7 era [24]. Similar trends have been observed in Europe and North America, driven by vaccine-induced serotype replacement and widespread antibiotic use [25,26]. The lower proportion of ST320 in our cohort may reflect differences in vaccine coverage, antibiotic pressure, or circulating serotypes. Nevertheless, even at 15%, its presence is clinically significant due to its association with multidrug resistance and potential for rapid clonal spread.

Globally distributed and clinically significant serotype 19A was the second most common serotype identified in present study. This serotype is notable for its high virulence and antibiotic resistance. In addition to the well-known MDR clone ST320/19A detected among our isolates, serotype 19A exhibited the highest number of STs, including ST199. Interestingly, ST199 was also associated with serotype 18A.

Changes in genes responsible for capsule biosynthesis can result in capsular switching, allowing ST199 to express different serotypes such as 19A and 18A. ST199 is a well-documented example of a globally distributed lineage that has undergone capsular switching, often following the introduction of PCV7 or PCV10 [27].

Among the 20 *S. pneumoniae* isolates with reduced susceptibility to penicillin that underwent MLST analysis, three were identified as serotype 3. This predominant serotype in our cohort is known for its high morbidity and mortality, particularly among older adults. Its exceptionally thick, mucoid capsule allows serotype 3 to evade immune defenses, leading to severe and persistent disease [28]. ST1377, a sequence type within the serotype 3 group, is presumed to share this high invasive potential. Notably, two of the three serotype 3 isolates in our study were associated with ST1377, suggesting potential clonal relatedness. However, the overall resistance burden of ST1377 is less frequently reported compared to other serotype 3 lineages [29]. In our study, ST1377 isolates exhibited reduced susceptibility to penicillin but remained susceptible to ceftriaxone, meropenem, and other commonly used antibiotics.

Despite widespread PCV13 use, serotype 3 continues to circulate globally, particularly among older adults, with persistent incidence in some regions attributed to ST1377 [30,31]. ST1377 has also been frequently detected in regions neighboring Croatia—in a surveillance study of IPD in Serbia, it was identified as one of the most common sequence types among serotype 3 isolates [32]. Similarly, in central Slovenia, ST1377 was among the most prevalent genotypes observed in IPD isolates [33].

The global Pneumococcal Serotype Replacement and Distribution Estimation (PSERENADE) project, which evaluates the impact of PCV10 and PCV13 on IPD incidence, reported that in countries using PCV10 in their national immunization programs, serotype 3 remains among the most common serotypes, while serotype 19A is the leading cause of IPD in adults over 50 years of age [34].

In our study, *S. pneumoniae* serotype 4/ST205 was identified in two cases. This lineage is well-characterized and primarily associated with invasive disease in adults. Notably, ST205/serotype 4 isolates were susceptible to most non-β-lactam antibiotics.

### 3.4. Non-β-Lactam Susceptibility

Beside penicillin resistance, the high rate of macrolide resistance in *S. pneumoniae* is a major global concern. Consequently, macrolide monotherapy is not recommended for empirical treatment of pneumococcal infections. Across Europe, macrolide resistance varies widely, but rates of approximately 20–28% are commonly reported among invasive isolates. The resistance rates observed in our study—particularly for macrolides (17.7%), tetracycline (15.2%), and trimethoprim-sulfamethoxazole (20%)—indicate moderate to relatively high selective pressure in our region. In countries where both β-lactam and macrolide resistance rates are high, fluoroquinolone resistance tends to be more frequent, although typically at lower levels [35]. In line with this, we observed fluoroquinolone resistance in 5.9% of invasive *S. pneumoniae* isolates.

All isolates in our study remained susceptible to vancomycin, which is reassuring and consistent with global data. Despite its pharmacological limitations related to tissue penetration and toxicity, vancomycin remains a reliable therapeutic option for severe infections or multidrug-resistant pneumococcal strains.

MDR *S. pneumoniae* typically shows reduced susceptibility to β-lactams, macrolides, tetracyclines, and sulphonamides, while resistance to quinolones is less frequent. Across Europe, the prevalence of MDR *S. pneumoniae* strains varies considerably—reaching up to 50% among carriage isolates in Spain, but as low as 4% in Sweden [8,9]. Surveillance of penicillin non-wild-type isolates in 2023 revealed significant inter-country variation across the EU/EEA, with some countries reporting MDR rates of ≥25% [36]. In our study, 15.3% of all *S. pneumoniae* isolates were identified as MDR. Notably, almost half of the isolates with reduced susceptibility to penicillin were MDR. A common resistotype among MDR isolates included resistance to erythromycin, clindamycin, and tetracycline. All three ST320 isolates, associated with serotype 19A, exhibited this MDR resistotype. Sequence types ST320, ST1830, ST271, ST319, and ST4467 were resistant to nearly all non-β-lactam antibiotic classes, with the exception of vancomycin.

### 3.5. Vaccination and Public Health Implications

Many European countries recommend pneumococcal vaccination for older adults as part of national strategies to prevent IPD. However, even where such policies exist, coverage remains low, below 30% in all countries [36]. In the present study, none of the patients had received pneumococcal vaccination despite national recommendations. Similarly, a study from France reported that most individuals had not been previously vaccinated against *S. pneumoniae* [11]. Adult immunization, particularly among older adults and those with underlying health conditions, remains a significant public health challenge. Improving vaccine coverage could reduce the burden of IPD in this vulnerable population. Notably, only 15.3% of IPD cases in our study were caused by serotypes not covered by PCV13 and PPV23—the vaccines recommended for individuals aged 65 years and older.

### 3.6. Study Limitations

This study has several limitations. First, its retrospective, single-center design limited the sample size, and the findings may not fully reflect the national serotype distribution in Croatia. Expanding surveillance to include additional hospitals across the country could provide a more comprehensive view of national trends. Second, focusing exclusively on invasive isolates may have introduced a selection bias toward more resistant clones, potentially overestimating antimicrobial resistance rates compared with broader community-based surveillance. Finally, only a subset of isolates with reduced penicillin susceptibility underwent MLST analysis, which may limit the completeness of our molecular epidemiological assessment. Future studies incorporating MLST or whole-genome sequencing of all isolates would provide a more detailed understanding of *S. pneumoniae* population structure and transmission dynamics in the region.

## 4. Materials and Methods

### 4.1. Study Setting

Surveillance was conducted from January 2022 to May 2025 in western Croatia at the Clinical Hospital Centre Rijeka, tertiary care academic institution with 1069 beds. All *S. pneumoniae* isolates obtained from patients aged 18 years and older were monitored in the Department of Clinical Microbiology. Clinical and demographic data were retrieved from the hospital’s electronic database.

Invasive isolates were defined as strains collected from blood, CSF, pleural fluid and other normally sterile sites. A single isolate per patient was studied. The total number of invasive pneumococcal isolates collected from inpatients and outpatients from Primorje-Gorski Kotar County, Croatia was 85. A total of 20 invasive *S. pneumoniae* isolates collected in 2024 were further subjected to MLST due to their reduced susceptibility to penicillin. Reduced susceptibility to penicillin was defined as isolates categorized as susceptible, increased exposure (I) or resistant (R), according to the guidelines of the European Committee on Antimicrobial Susceptibility Testing (EUCAST), 2025 [37].

Mortality was assessed based solely on in-hospital outcomes.

### 4.2. Antibiotic Susceptibility Testing and Serotyping

*S. pneumoniae* isolates were identified using standard microbiological methods, including optochin susceptibility testing and the VITEK^®^2 system (bioMérieux, Marcy-l’Étoile, France). Antimicrobial susceptibility to penicillin, clindamycin, erythromycin, moxifloxacin, tetracycline, trimethoprim/sulfamethoxazole, and vancomycin was determined by disk diffusion (Liofilchem, Roseto degli Abruzzi, Italy) in accordance with the EUCAST 2025 guidelines [37]. As the interpretation of penicillin susceptibility depends on the type of clinical specimen (i.e., meningitis vs. non-meningeal infections), meningitis breakpoints were applied for CSF isolates. The MICs of penicillin, ceftriaxone and meropenem were determined using MIC Test Strips (Liofilchem, Roseto degli Abruzzi, Italy) and the broth microdilution Micronaut system (Merlin Diagnostika GmbH, Bornheim, Germany). Isolates exhibiting resistance to at least one agent in three or more antimicrobial classes were classified as MDR.

Serotyping was performed at the Croatian Pneumococcal Reference Laboratory for *S. pneumoniae* by Quellung reaction with pool/group/type/factor-specific commercial antisera (Statens Serum Institute, Copenhagen, Denmark).

### 4.3. DNA Extraction and MLST

A total of 20 invasive *S. pneumoniae* isolates with reduced susceptibility to penicillin were subjected to MLST. Before DNA extraction, bacterial pellet was resuspended in a Buffer BL2 (SaMagBacterial DNA Extraction Kit, Sacace Biotechnologies, Como, Italy) designed for bacterial cell wall lysis. Briefly, a loop of bacterial colonies from an overnight culture grown on blood agar were placed in 200 µL of Buffer BL2 (SaMagBacterial DNA Extraction Kit, Sacace Biotechnologies, Como, Italy) and heated at 70 °C for 1 h. Genomic DNA was then extracted using the SaMagBacterial DNA Extraction kit and the SaMag-12^TM^ automatic nucleic acids extraction system (Sacace Biotechnologies, Como, Italy). The extracted DNA was quantified using the Qubit™ dsDNA HS Assay Kit (Thermo Fisher Scientific, Waltham, MA, USA) and stored at −20 °C until use. The PCR amplification of the seven housekeeping genes (*aroE*, *gdh*, *gki*, *recP*, *spi*, *xpt*, and *ddl*) was conducted using the primers as specified at https://pubmlst.org/organisms/streptococcus-pneumoniae/primers, accessed on 12 November 2025 [38]. The PCR product of housekeeping genes was purified with a QIAquick PCR Purification Kit (Qiagen, Germantown, MD, USA) according to the supplied instructions. To verify the sizes of the PCR products, agarose gel electrophoresis was performed. PCR products were subsequently subjected to Sanger sequencing at a commercial facility (Macrogen Europe B.V., Amsterdam, The Netherlands) The sequences of the seven housekeeping genes were trimmed, aligned, and processed using Codon Code Aligner software (v 12.0.4, CodonCode Corporation, Dedham, MA, USA), a standalone program [39]. Sequence types (STs) were assigned by submitting the allele sequences to the PubMLST database (https://pubmlst.org/organisms/streptococcus-pneumoniae, accessed on 12 November 2025) [40]. Correlations between STs and serotypes were determined based on PubMLST data and supporting literature [32,33,40].

### 4.4. Statistical Analysis

Data were analyzed using Statistica 14.0.0 (TIBCO Software Inc., 2020, Palo Alto, CA, USA). Categorical variables were compared using the Chi-square test and Two-sample proportion Z-test, and continuous variables using Student’s *t*-test or the Mann–Whitney U test, as appropriate. Continuous variables are presented as means, and categorical variables as numbers and percentages. A *p*-value of <0.05 was considered statistically significant.

## 5. Conclusions

This study highlights the predominance of serotypes 3 and 19A, particularly the multidrug-resistant ST320 clone, among invasive *S. pneumoniae* isolates in Croatia. The rising penicillin non-susceptibility and notable rates of macrolide and tetracycline resistance emphasize the need for sustained antimicrobial surveillance and prudent antibiotic use. Broader pneumococcal vaccine coverage in adults and continued molecular monitoring are essential to mitigate the burden of invasive pneumococcal disease.

## Figures and Tables

**Figure 1 antibiotics-14-01158-f001:**
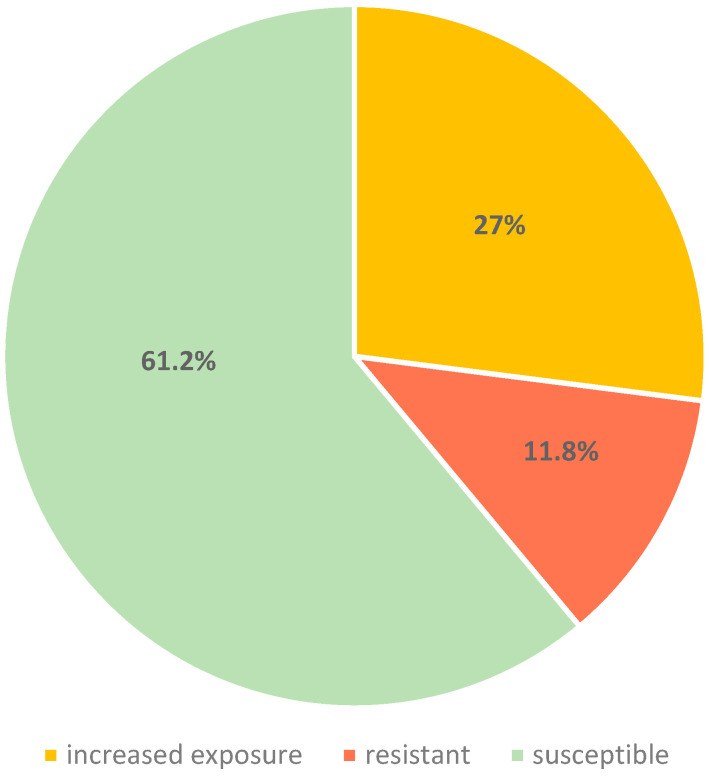
Susceptibility to penicillin among invasive *S. pneumoniae* isolates, 2022–2025.

**Figure 2 antibiotics-14-01158-f002:**
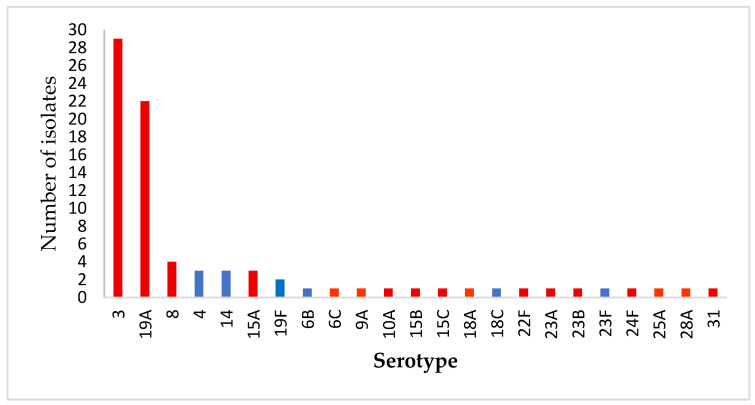
Distribution of all *S. pneumoniae* serotypes. Blue bars: PCV10 serotypes; red bars: serotypes not included in PCV10.

**Figure 3 antibiotics-14-01158-f003:**
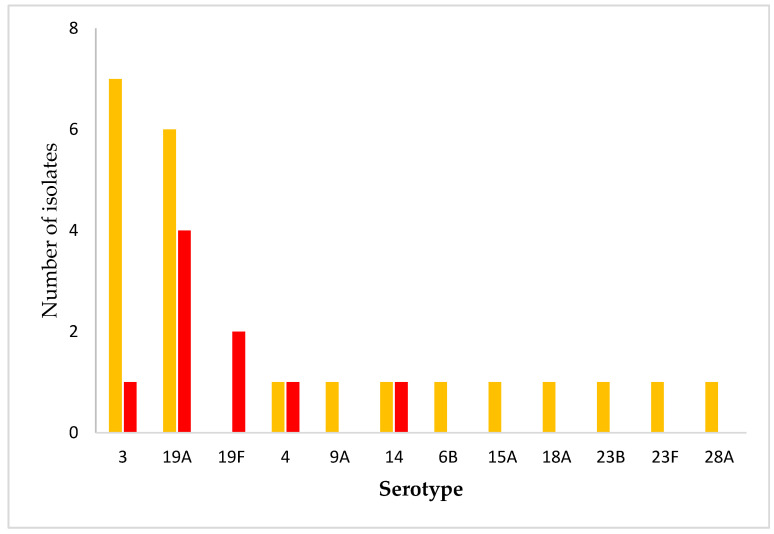
Distribution of *S. pneumoniae* serotypes with reduced penicillin susceptibility. Orange bars: susceptible, increased exposure (I); red bars: resistant (R).

**Figure 4 antibiotics-14-01158-f004:**
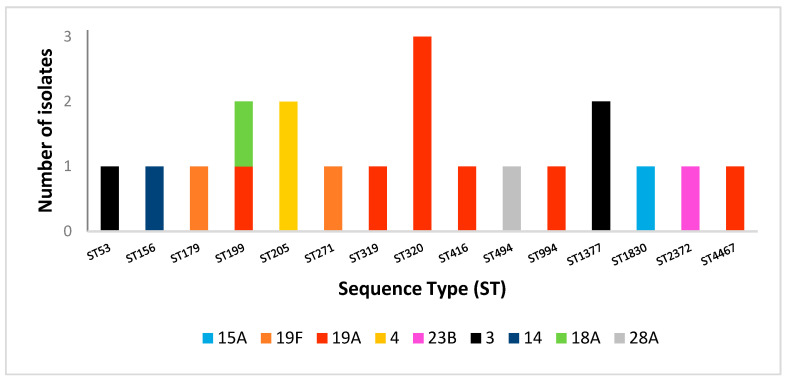
Distribution of sequence types (STs) of *S. pneumoniae* isolates with reduced susceptibility to penicillin and their corresponding serotypes.

**Table 1 antibiotics-14-01158-t001:** Number of *S. pneumoniae* isolates by age groups.

Age Group	*n* (%)
18–35	2 (2.4)
36–49	7 (8.2)
50–64	25 (29.4)
≥65	51 (60)

**Table 2 antibiotics-14-01158-t002:** Clinical-demographic characteristics of IPD patients.

	All Patients (*n* = 85)	*p*
^a^ Age, mean ± SD	68.4 ± 15.7	
^b^ Gender		0.173
Female, *n* (%)	39 (45.9)	
Men, *n* (%)	46 (54.1)	
^c^ Underlying conditions		<0.001 *
No comorbidities, *n* (%)	9 (10.6)	
1	21 (24.7)	
≥2	55 (64.7)	
ICU admission, *n* (%)	33 (38.8)	
CCI median (IQR)	4 (3–6)	
^b^ Mortality, R *n* (%)	25 (29.4)	<0.001 *
^c^ Culture Site		<0.001 *
Blood, *n* (%)	74 (87.1)	
CSF, *n* (%)	6 (7.1)	
Other, *n* (%)	5 (5.8)	

Abbreviations: ICU, intensive care unit; IPD, invasive pneumococcal disease; IQR, interquartile range; *n*, number; SD, standard deviation; * statistical significance; CCI, Charlson Comorbidity Index; CSF, Cerebrospinal Fluid ^a^ Student’s *t*-test; ^b^ Two-sample proportion Z-test; ^c^ Chi-square test.

**Table 3 antibiotics-14-01158-t003:** Comparison of clinical and demographic characteristics between surviving and deceased patients with invasive pneumococcal disease.

	Survived (*n* = 60)	Deceased (*n* = 25)	*p*
^a^ Age, mean ± SD	66.1 ± 16	74.2 ± 13.5	0.029 *
^b^ Gender			0.591
Female, *n* (%)	29 (74.4)	10 (25.6)	
Men, *n* (%)	30 (66.7)	15 (33.3)	
^b^ Underlying conditions			0.07
No comorbidities, *n* (%)	9 (15)	0 (0)	
1, *n* (%)	17 (28.3)	4 (16)	
≥2, *n* (%)	34 (56.7)	21 (84)	
ICU admission			<0.001 *
Yes, *n* (%)	16 (26.7)	17 (68)	
No, *n* (%)	44 (73.3)	8 (32)	
^c^ CCI median (IQR)	4 (2–5)	5 (3–7)	0.010 *

Abbreviations: ICU, intensive care unit; IQR, interquartile range; *n*, number; SD, standard deviation; * statistical significance; CCI, Charlson Comorbidity Index; ^a^ Student’s *t*-test; ^b^ Chi-square test; ^c^ Mann–Whitney U test.

**Table 4 antibiotics-14-01158-t004:** MIC_90_ values and susceptibility of invasive *S. pneumoniae* isolates to β-lactam classes of antibiotics.

Antibiotic	MIC_90_ mg/L	I *n* (%)	R *n* (%)
Penicillin	0.75	23 (27)	10 (11.8)
Ceftriaxone	0.5	4 (4.7)	4 (4.7)
Meropenem	0.094	0 (0)	1 (1.2)

Abbreviations: I, increased exposure; MIC, minimum inhibitory concentration; R, resistant.

**Table 5 antibiotics-14-01158-t005:** Susceptibility of invasive *S. pneumoniae* isolates to non-β-lactam classes of antibiotics.

Antibiotic	R (%)
Erythromycin	15 (17.7)
Moxifloxacin	5 (5.9)
Tetracycline	13 (15.2)
Trimethoprim-sulfamethoxazole	17 (20)
Vancomycin	0

Abbreviations: R, resistant.

**Table 6 antibiotics-14-01158-t006:** Distribution of *S. pneumoniae* isolates with reduced susceptibility to penicillin according to sequence types (STs), serotypes, and antimicrobial susceptibility patterns.

Isolates Number	Serotype	Antimicrobial Susceptibility					MDR	Sequence Type
		ERY	CLI	MOX	SXT	TE		
1	15A	R	R	R	S	R	yes	1830
2	19F	R	R	R	I	R	yes	179
3	19A	R	R	R	R	R	yes	320
4	19A	R	R	S	R	R	yes	320
5	19A	R	R	S	R	R	yes	320
6	4	S	S	S	S	S	no	205
7	19A	S	S	S	S	S	no	199
8	19F	R	R	S	R	R	yes	271
9	23B	S	S	S	R	S	no	2372
10	3	S	S	S	S	S	no	1377
11	4	R	R	S	S	S	no	205
12	19A	S	S	S	S	S	no	994
13	14	S	S	S	R	S	no	156
14	3	S	S	S	S	S	no	53
15	18A	S	S	S	S	S	no	199
16	28A	S	S	S	S	S	no	494
17	3	S	S	S	S	S	no	1377
18	19A	R	R	S	R	R	yes	319
19	19A	R	R	S	S	R	yes	416
20	19A	R	R	S	R	R	yes	4467

Abbreviations: CLI, clindamycin; ERY, erythromycin; MOX, moxifloxacin; SXT, trimethoprim-sulfamethoxazole; TE, tetracycline.

## Data Availability

Data supporting this article is available from the corresponding author upon reasonable request.

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
