# Peer review of "Molecular Epidemiology, Antimicrobial Resistance, and Clinical Characteristics of Streptococcus pneumoniae Isolated from Adult Patients with Invasive Pneumococcal Disease"

_antibiotics, 2025, doi:10.3390/antibiotics14111158_

Round 1
Reviewer 1 Report
Comments and Suggestions for Authors
The paper describes the molecular characterization of invasive pneumococcal disease (IPD) isolates from adults in western Croatia between 2022 to 2025.
The total number of strains over a period of four years was 85, of which only 20 were determined to be ST. This study identified a limited number of STs, which does not fully show the molecular epidemiological characteristics in the region. All 83 strains should be classified by ST.
In this study for MIC, the guidelines of the European Committee on Antimicrobial Susceptibility Testing (EUCAST) 2025. Regarding penicillin resistance, various notations are used as follows.
Line 77, reduced susceptibility to penicillin (PRSP);Line 156, increased exposure penicillin susceptibility (I category); Line 158, penicillin-resistant isolates (R category); In Figure 3, reduced penicillin susceptibility (PRSP),Orange bars: increased exposure; red bars: resistant.
These notations should be standardized. The EUCAST guidelines only include categories S and R, but what is the basis for the MIC levels in Category “I”?
However, for pneumococcal MICs, three categories are commonly used: S (Susceptible), I (Intermediately susceptible), or R (Resistant) using the break points stipulated in CLSI (Clinical and Laboratory Standards Institute. In the CLSI, S. pneumoniae (SP) with Penicillin MICs ≤0.06 μg/mL was classified as penicillin susceptible SP (PSSP), that with MIC ≥ 0.125 μg/mL; ≤1.0 μg/mL as penicillin intermediate resistant SP (PISP) and that with MIC ≥ 2 μg/mL as penicillin resistant SP (PRSP).
Reviewer 2 Report
Comments and Suggestions for Authors
This study provides significant data on the molecular features and resistance patterns of S. pneumoniae in adult patients. Overall, the methodology is appropriate, and this manuscript effectively combines and presents molecular and clinical data. However, minor revisions should be performed in the manuscript.
L25: "Serotypes" should be written as "The serotypes"
There is too much similarity between the words or scope of the first sentences of Background/Objectives, Line 18, and Introduction, Line 44. This situation provides for a disconnected flow of information.
The term "normal flora" is now considered outdated and no longer a scientifically accurate definition. It should be replaced with the most recent, more accurate academic term, such as "microbiome of the upper respiratory tract" or another term.
The use of the general term "pneumococcus" (Line 63) is common. However, it could be replaced by the entire scientific species denomination, "S. pneumoniae," throughout the manuscript.
Line 70: The exact number of housekeeping genes considered in the analysis, rather than using the general term "several," should be specified.
Line 88: The exact p-value should be reported along with the findings. Thus, the evaluation of the result would be more meaningful statistically.
Line 121: "Antimicrobial susceptibility" should be written as "Antimicrobial susceptibility testing"
Line 168: The criteria or reference that was used to establish the agreement or correlation between the defined STs and serotypes should be presented.
Figure 4 should be located to the right. It should be appropriately centered or aligned.
Line 185: Since PRSP has been defined earlier in the text, the complete expansion of the abbreviation is not required.
The discussion
Line 189: A few sentences about the subject of the manuscript should be given.
Line 191-193: This paragraph is unnecessary and should be rewritten to introduce.
Line 427: The name of the buffer should be specified.
Line 443, A proper citation for the software called "Codon Code Aligner" mentioned, indicating whether it is a user interface, a standalone program, or an online tool, should be given.
Round 2
Reviewer 1 Report
Comments and Suggestions for Authors
Depending on the region and serotype, same genetic lineage may be observed between penicillin-sensitive strains and penicillin-resistant strains. We would like you to investigate that point as well next time.